# Engineering Learning Outcomes: The Possible Balance between the Passion and the Profession

**Diana Dias** [1,2]

1    CIPES—Centre for Research in Higher Education Policies, 4450-227 Matosinhos, Portugal;
     diana.dias@ulusofona.pt
2    Lusofona University, 1749-024 Lisboa, Portugal

**Abstract:** What is an engineering student expected to learn to become a competent engineer? Which is most desirable in higher education: hard skills or soft skills? Are there differences between master's and bachelor's learning outcomes, or between public or private schools? Previous works point out the relevance of hard rather than soft skills in engineering higher education. The implementation of learning outcomes (LOs) in higher-education curricula has been a common challenge for European educational institutions. Despite the efforts undertaken since the Bologna Declaration, the effective implementation of the learning outcomes paradigm is still in process. This research intends to analyse the LOs proposed in the scope of graduate and postgraduate electrical or computer electrical or computer engineering programs submitted to the Agency for Assessment and Accreditation of Higher Education (A3ES) in Portugal since 2004. Conducting a qualitative analysis, data documents were coded according to the Nusche typology of LOs. The results suggest that cognitive skills are a core dimension in electrical or computer engineering LOs. References to non-cognitive LOs are not representative. Different institution sectors (private vs. public), systems (universities vs. polytechnics), and study cycles (first vs. second vs. integrated master) highlight distinct cognitive and non-cognitive categories in their definition of LOs. The results are discussed in terms of a changing or a recycling paradigm in higher-education practices.

**Keywords:** learning outcomes; engineering; higher education; cognitive learning outcomes





## 1. Introduction

Higher education has undergone considerable changes in recent years, as an answer to Bologna challenges (Dias and Amaral 2014). Portugal is not an exception. Higher education institutions (HEIs) have been challenged to undertake an accreditation of their full programme portfolio. This request converges to a re-conceptualization of the educational offer of higher education, which could represent an opportunity to reflect on how the training objectives associated with higher education are perceived and conceptualized by the different HEIs (Santiago et al. 2014).

Accordingly, this paper aims to evaluate the level of implementation of LOs by HEIs in Portugal, specifically on electrical or computer engineering. To this end, an analysis of international educational politics, emphasizing European context and the Portuguese "National Qualification Framework" guidelines will be carried out. The adoption of a qualitative approach enables the extrication of dissimilar conceptualizations of LOs, differing not only on the subsystems (universities versus polytechnics) but also on the institution sectors (private versus public), and on the qualification level (first cycle, integrated master, and second cycle). The findings are used to assess the level(s) and method(s) of LO implementation in the Portuguese higher education system, emphasizing how electrical or computer engineering LOs are being defined by academia.

## 2. Learning Outcomes: A Brief History with Extended Implications

One of the major goals of the Bologna process is to endorse European cooperation in the promotion of quality assurance in higher education systems, making them comparable not only in what concerns the analysis criteria but also in the implementation and evaluation methods (Dias and Amaral 2014). The definition of a common "Framework for Qualification of European Higher Education" to be adopted by the European countries is a clear example of the comparability and transparency request (Veiga et al. 2013). Qualification frameworks are crucial tools for a clear and operational description of what a student is expected to learn during the acquisition of a particular qualification (Dias and Amaral 2014). Linked to these descriptors and frameworks, LOs became an important approach in educational policies in Europe (Stanley 2015). Adopting the definition proposed by Adam (2008) or Scott (2011), LOs are statements of what a student should know, understand and/or be able to demonstrate at the end of a learning period, describing what a learner will have learnt throughout such a period. In this sense, LOs are described as "a fundamental building block of the Bologna educational reform" (Adam 2006, p. 3).

European educational systems must assure their curricula comparability as regards structure, study programmes, and pedagogical methodologies. Henceforward, assessment tools, curricula design practices, scientific contents, and LOs need to be brought into line, to grant a link not only between programmes but also between European higher education institutions (Dias and Amaral 2014).

As stressed by Adam (2004), LOs are mostly defined in terms of knowledge, skills, abilities, attitudes, and the understanding that the student will benefit because of a learning experience. LO assessment assumes particular significance in the scope of the 2030 Agenda for Sustainable Development (UNESCO 2017; Vidal 2019), considering the Sustainable Development Goals (SDG) to be achieved within a plan of action for people, planet, and prosperity. Education, addressed in SDG 4, i.e., "Ensure inclusive and equitable quality education and promote lifelong learning opportunities for all", is the key pillar of the 2030 Agenda and had direct repercussions for the remaining goals. So, SDG 4, focusing on education, needs to be successfully monitored to assess the results obtained in fulfilling an education with quality in the curricula of all disciplines.

LOs emphasise what a student is expected to acquire regarding knowledge, skills, and competences, according to each qualification level (Wiliam 2010; Gallavara et al. 2008; Tissot 2014). In other words, LOs must not only describe the expected or intended learning outputs but should also point out how these achievements will be assessed. So, LOs help students to recognize in advance what they are expected to know, understand, and accomplish, whether for a given programme or for a specific class, as well as the assessment criteria that will be used. Hence, LOs can be differentiated from competencies. In fact, competencies and outcomes can both be written to describe the learning gained by students in a learning period, but they do not mean the same thing. Hartel and Foegeding (2004) present competency as a general statement that describes the desired knowledge, skills, and behaviours of a student graduating from a program. Competencies usually describe the applied skills and knowledge that enable students to successfully perform in professional, educational, and other life contexts. In turn, LOs are very detailed statements that explain precisely what a student will be able to do in some measurable way. For a given competency, there may be more than one measurable outcome defined. Thus, a true learning outcome must be written so that it can be measured or assessed. While the term 'LO' is typically applied in the context of a program or course, the term 'competency' is more commonly employed with regard to professional fields (i.e., engineering, psychology).

However, the descriptions and understandings of LOs are not consensual, embracing a wide range of typologies and categories (Dias and Soares 2017; Scott 2011). The Assessment of Higher Education Learning Outcomes (AHELO), an OECD study, defined LOs as a combination of generic skills and discipline-specific skills (Dias and Amaral 2014). Critical thinking, discipline knowledge, problem-solving, teamwork, communication, professional skills, ethics and values, creativity, and learning to learn are some examples of

the most-mentioned LOs in that study. To González and Wagenaar (2003), three types of generic competencies could be distinguished: instrumental competences refer to cognitive, methodological, technological, and linguistic abilities; interpersonal competencies integrate individual abilities such as social skills; and systemic competencies encompass abilities and skills concerning whole systems. Tuning Project regards a list of 31 generic competences (González and Wagenaar 2003), some of them linked to the ability to communicate in a second language, the capacity to learn and stay up-to-date with learning, the ability to communicate both orally and through the written word in a first language, the ability to be critical and self-critical, the ability to plan and manage time, the ability to reveal awareness of equal opportunities and gender issues, the capacity to generate new ideas, the ability to search for, process, and analyse information, the ability to identify, pose and solve problems, the ability to make reasoned decisions, the ability to undertake research at an appropriate level, the ability to work in a team or in an international context, the ability to act based on ethical reasoning, or the spirit of enterprise and the ability to take initiative. In line with this, Adam (2006, 2008) advocates that LOs are a "mixture of knowledge, skills, abilities, attitudes and understandings that an individual will achieve" (p. 2), divided between subject-specific (i.e., related to the subject discipline) and generic (i.e., key transferable skills related to all disciplines) outcomes. Nusche (2008) defends a divergent perspective on LO typologies: "What a learner knows or can do as a result of learning" (p. 7) could be categorized in terms of cognitive and non-cognitive outcomes. For Bloom (1956), education should go further than simple factual knowledge and understanding. Academic outcomes such as practical application, synthesis, analysis, and assessment should be considered learning goals.

## 3. Categorizing Learning Outcomes

Even considering the multiplicity of learning outcome definitions (OECD 2013), it is possible to outline some convergence points between perspectives. For example, Nusche (2008) recommends a categorization of learning outcomes for higher education that encompasses a set of guidelines, seeking to identify the different dimensions of learning, as well as their assessment methods. As previously stated, the typology proposed by Nusche comprises cognitive and non-cognitive outcomes.

### 3.1. Cognitive Learning Outcomes

Learning on a cognitive level refers to the ability to access memory and retrieve past knowledge as well as the development of intellectual amplitude and skills (Strike and Posner 1992). Cognitive LOs range from the most specific areas of knowledge to the broader processes of thinking and problem-solving (Shavelson and Huang 2003). A range of classifications of cognitive LOs has been proposed in the literature (for further reference: Eraut 2004; Gagné 1977; Klemp 2001; Kolb 1981; Marzano 2001), based mainly on Bloom's (1956) taxonomy of learning objectives.

Focusing on Nusche's (2008) perspective, cognitive LOs are differentiated in terms of knowledge and skill acquirements. LOs in terms of knowledge focus on a general field of know-how or a specific scientific domain. General knowledge refers to a basic curriculum considered as "essential learning" (Maeroff 2006), culturally basic and independent from their scientific area (Quintana et al. 2016). Accordingly, as already suggested by Vygotsky (1978) "learning is a necessary and universal aspect of the process of developing culturally organized, specifically human and psychological function" (p. 90). In turn, specific knowledge refers to a specific scientific topic such as psychology, engineering or architecture. Evaluating specific knowledge may prove inestimable when comparing the depth of a certain theme throughout curricular units, as well as the educational quality between subject matters, programmes, or higher education institutions. In terms of skills acquirement, cognitive LOs integrate, for instance, verbal and quantitative thinking, understanding, analytical processing, critical thinking or problem solving, i.e., skills that are transferable across different disciplinary subjects.

Cognitive skills are based on complex processes of thought, such as verbal and quantitative thinking, understanding, analytical processing, critical thinking, problem-solving, and the evaluation of new ideas. According to Allen and Van der Velden (2005), such cognitive skills transcend a specific area of study, being transferable between themes, courses, and contexts. However, some skills are specific and dependent on an area of study. These specific skills may be perceived as a set of applied patterns of thought used in a more global scope, such as, for instance, social sciences or biology. Research methods, work procedures, or specific analysis methodologies on a given subject matter are examples of specific skills (Nusche 2008).

*3.2. Non-Cognitive Learning Outcomes*

Opposed to cognitive LOs, non-cognitive learning refers to changes in beliefs or the evolution of certain standards and civic values (Ewell 2005). The promotion of non-cognitive learning has been a core aim in higher education curricula (Billings and Terkla 2014). Psychosocial development encompasses self-development, namely, maturing identity and self-esteem growth, as well as building up relationships among peers, with teachers, and with the institutions they are inserted into. Interpersonal and intercultural know-how, as well as autonomy and maturity, are also connected with psychological development. In turn, behaviours and ethics, despite being intimately connected, are two distinct dimensions. Behaviours are beliefs directed towards a specific object, while ethics are general patterns of action that transcend behaviours (Pascarella and Terenzini 2005). Social awareness, learning motivations, and respect for diversity are examples of non-cognitive LOs concerning behaviours and ethics (Dias and Soares 2017; Reason and Hemer 2012). Institutions have an active role in the promotion of civic and psychosocial dimensions besides the promotion of knowledge and/or the acquisition of practical skills (Reason and Hemer 2012).

Non-cognitive LOs can be explored either in class or in extra-curricular activities, such as coaching, counselling, and school sports, or even through associateship (Nusche 2008). Pascarella and Terenzini (2005) compiled an excess of 1500 studies in the United States, trying to explore the impact of higher education on students. The most notable improvements are related to psychosocial development, behaviours, and ethics. Psychosocial development encompasses self-development, namely identity maturing and self-esteem growth, as well as building up relationships among peers, with teachers, and with the institution they are in. Interpersonal and intercultural know-how as well as autonomy and maturity are also connected with psychological development. In turn, behaviours, and ethics, despite being intimately connected, are two distinct dimensions. Behaviours are beliefs directed towards a specific object, while ethics are general patterns of action that transcend behaviours (Pascarella and Terenzini 2005). Social awareness, learning motivation, and the respect for diversity are examples of non-cognitive LOs concerning behaviours and ethics (Reason and Hemer 2012).

Considering the diversity of LOs, there is a recognition gap in those that are most valued by higher education institutions, particularly in the engineering area. Thus, this study aims to explore which LOs are specifically emphasised in study programmes submitted to quality accreditation in the Portuguese Agency for Assessment and Accreditation of Higher Education (A3ES), i.e., what knowledge, skills, competences, and attitudes are reinforced as expected LOs in the engineering education competency profiles. The identification of these LOs can reduce the gap identified by Dias et al. (2019) by determining whether these LOs meet the needs of future engineers.

**4. Methodology**

*4.1. Measures*

The implementation of LOs by Portuguese higher education institutions may be assessed by an analysis of the data those institutions submitted to the Agency for Evaluation and Accreditation of Higher Education (A3ES). At the initial stage of the accreditation

process, higher education institutions submit an accreditation proposal to A3ES entitled "Request for Prior Accreditation of a New Study Cycle". In the description of each study cycle that is submitted, institutions must describe what the "intended LOs" expected to be achieved by students at the end of a specific learning period are. Data analysis focused on information included in this specific question, which is limited by A3ES to 1000 characters. All reports submitted under the Request for Evaluation of New Study Cycles (NSC) to A3ES concerning electrical or computer engineering were collected (17 reports). A defined section of those documents was analysed, focusing on the question asking for a description of "intended LOs of the study cycle".

Both higher education subsystems (universities and polytechnics), both private and state sectors, and all HE degrees were represented, as can be analysed in the following table (Table 1):

**Table 1.** New Study Cycles in Electrical or Computer Engineering.

| HE | Degree | | | Subsystem | | Sector | |
|---|---|---|---|---|---|---|---|
| NCS | Master | Integrated Master | Bachelor | University | Polytechnic | Public | Private |
| N | 10 | 3 | 4 | 10 | 7 | 11 | 6 |
| % | 58.8 | 17.7 | 23.5 | 58.8 | 41.2 | 64.7 | 35.3 |

In Portugal, in higher education, undergraduate and postgraduate programs in electrical or computer engineering can be taught both in the university subsystem and in the polytechnic education subsystem. According to the law, the difference is that "university education is oriented towards offering solid scientific training, joining efforts and competences from teaching and research units, and polytechnic education focuses especially on vocational training and on training advanced techniques, professionally oriented". Thus, universities offer scientifically solid training, which combines skills from teaching and research units, being able to teach not only bachelor's (180 ECTS) but also master's (120 ECTS), as well as integrated master's (300 ECTS). Those who want to practice a profession, especially oriented towards their vocation and one that is technically advanced, tend to choose polytechnic education. As universities, polytechnic institutions can offer undergraduate and master's degrees, but not integrated master's degrees, which are offered exclusively by university education. Undergraduate degrees tend to offer more broad-band training but already specialized in electrical or computer engineering. Master's degrees are also focused on more specific specializations. Integrated master's are cycles of initial training studies taught only in university education, which have at least 300 ECTS credits and a normal extent of between 10 and 12 semesters, in cases where this duration for access to the exercise of a certain professional activity is established by legal norms of the European Union or results from a stable and consolidated practice in the European Union.

Ten reports correspond to new master's degrees, 4 to new bachelor's degrees, and 3 to integrated master's degrees. Of those 17 reports, 10 were submitted by universities, 7 of which were from the public sector. The distribution between subsystems is better balanced as there are almost as many universities as they are polytechnic institutions. The state sector has a higher incidence (61.5%) of master's degrees. Specifically, 23.1% correspond to integrated master's, while 15.4% correspond to bachelor's degrees. In the private sector, the reports analysed refer to 50% for each master's degrees and bachelor's degrees. Here, no integrated master's were submitted. Requests for new master's study cycles prevailed in the public sector, making up 80% of the total submissions. On the other hand, the number of requests for new bachelor's degrees is equivalent in both sectors. Considering the subsystem, 42.9% of university submissions are concerned with master's degrees as well as integrated master's degrees. Focusing on polytechnic subsystems, 70% of their submissions refer to master's degrees, while 30% are concerned with bachelor's degrees. Regarding the analysis of the sector, 90% of the total master's degrees and 50% of the total

bachelor's degrees are submitted by the public sector. Looking at the private sector, 66.7% refer to new bachelor's degree submissions while 33.3% refer to new master's degrees.

*4.2. Procedure*

To conduct this study, a qualitative approach was adopted. The prime goal of the research is to analyse how LOs are defined in Portuguese electrical or computer engineering curricula. The analysis was performed using the MaxQDA package (version 12), a software program designed for computer-assisted qualitative and mixed-methods data, text, and multimedia analysis. This tool enables coding data documents, allowing the analysis and exploration of relations between codes. The versatility and flexibility of this software enable a more complete analysis of qualitative data, highlighting semantic information (rather than syntax information). Data analysis was carried out in two steps. Firstly, a search for the most relevant words provided a general overview of the content underlying LOs, adopting a summative content analysis that involves counting and comparisons, usually of keywords or content, followed by the interpretation of the underlying context. Secondly, a directed content analysis was performed. A deductive approach to qualitative analysis was selected. The Nusche theory was used as a starting point and the data was used to either support or build upon that framework. Using existing theory or prior research allowed us to begin by identifying key concepts or variables as initial coding categories. Next, operational definitions for each category were determined using the theory. The transcriptions were arranged according to their themes and were analysed using the constant comparative method (Glaser 1992). The analysis started by grouping the transcriptions according to the categories suggested by Nusche (2008): cognitive LOs (specific and general knowledge/specific and general skill) and non-cognitive LOs (behaviours/ethics/psychosocial development).

## 5. Results

Using a summative content analysis, the analysis of word frequency in the documents submitted for accreditation indicates that "skills", "ability" (N = 23), and "knowledge" (N = 22) are the most mentioned terms. Other words, such as "management" (N = 20), "engineering" (N = 17), "systems" (N = 15), "development" (N = 14), and "problems" (N = 13), were also stressed. The words correspond to 13.4% of the total words studied. Figure 1 represent graphically the analysis of word frequency on LOs mentioned in Portuguese electrical or computer engineering curricula of new study cycles submitted for accreditation.

Considering the directed content analysis, in a deductive basis, with Nusche theory was used as a reference to the analysis, 140 excerpts were extracted, which, on average, corresponds to 8.24 citations per report. On average, each submission for "Request for Evaluation of New Study Cycles" has about 8 different references to LOs.

As shown in Table 2, 77.8% were tagged as skills, and the remaining 22.02% were tagged as knowledge.

Within the skills, 56.5% were tagged as specific skills, with the remaining 43.5% identified as general skills.

Concerning knowledge, 87.5% were tagged as specific knowledge while 12.5% were of a more general nature.

When dealing with non-cognitive LOs, which make up for 22.1% of the total LOs, within those, behaviours plus ethics and psychosocial development have a distribution of 54.8% and 45.2%.

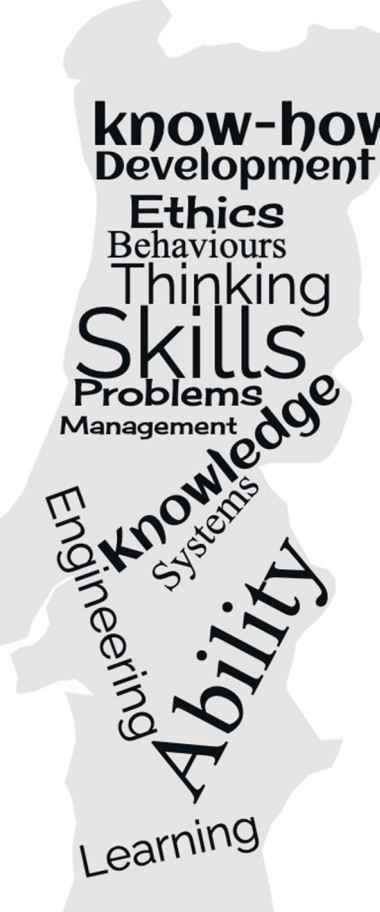

**Figure 1.** Analysis of word frequency on LOs mentioned in Portuguese electrical or computer engineering curricula of new study cycles submitted for accreditation.

**Table 2.** Percentage of references for each category.

| | | |
|---|---|---|
| **Cognitive LOs (77.9%)** | Skill (77.08%) | Specific skills (56.5%) |
| | | Generic skills (43.5%) |
| | Knowledge (22.02%) | Specific knowledge (87.5%) |
| | | General knowledge (12.5%) |
| **Non-cognitive LOs (22.1%)** | Behaviours (54.8%) | |
| | Ethics and psychosocial development (45.2%) | |

When contrasting higher education sectors, public institutions have, on average, 10.2 LOs per report, while the private sector has an average of 7. As displayed in Table 3, cognitive LOs are well represented both in public and private sectors. However, while cognitive skills are equally mentioned by both sectors, cognitive knowledge is preferably referred to by the public sector. Focusing on cognitive skills, the private sector has a superior average on specific skills. On the other hand, no single tag for generic skills in the private sector is observed. A similar pattern is observed in knowledge outcomes. Specific knowledge is mostly mentioned by the private sector, whereas generic knowledge is referred to only by public institutions. Regarding non-cognitive LOs, public institutions show better results than the private sector (2.14 average against just 0.33). LOs concerning or associated with psychosocial development as well as behaviours and ethical values are mentioned at least once per report.

**Table 3.** Incidences (average) by IES sector.

| | | |
|---|---|---|
| **Cognitive LOs**<br>**Public: 8.1 \| Private: 6.7** | Skill<br>Public: 5.4 \| Private: 3.3 | Specific skills<br>Public: 2.7 \| Private: 3.33 |
| | | Generic skills<br>Public: 2.6 \| Private: 0 |
| | Knowledge<br>Public:1.36 \| Private: 1.7 | Specific knowledge<br>Public: 1.1 \| Private: 1.7 |
| | | General knowledge<br>Public: 0.2 \| Private: 0 |
| **Non-cognitive LOs**<br>**Public: 2.14 \| Private: 0.33** | Behaviours and ethics<br>Public: 1.44 \| Private: 0.33 | |
| | Psychosocial development<br>Public: 1 \| Private: 0 | |

Table 4 presents results for each category, distinguishing between university and polytechnic systems. Cognitive LOs, mainly knowledge outcomes, are mostly stated by universities. Nonetheless, polytechnics refer to cognitive skills more often than universities. When refining the data, the results indicate that generic skills are mentioned more regularly by universities, while specific skills are more often referred to by polytechnics. Both universities and polytechnics have similar results in non-cognitive LOs.

**Table 4.** Incidences (average) by system.

| | | |
|---|---|---|
| **Cognitive LOs**<br>**University: 6.9**<br>**Polytechnic: 6.1** | Skill<br>University: 4.6<br>Polytechnic: 5.3 | Specific skills<br>University: 1.7 \| Polytechnic: 3.6 |
| | | Generic skills<br>University: 2.9 \| Polytechnic: 1.7 |
| | Knowledge<br>University: 2.3 Polytechnic: 0.8 | Specific knowledge<br>University: 1.9 \| Polytechnic: 0.8 |
| | | General knowledge<br>University: 0.43 \| Polytechnic: 0 |
| **Non-cognitive LOs**<br>**University: 1.7**<br>**Polytechnic: 1.9** | Behaviours and ethics<br>University: 1 \| Polytechnic: 1 | |
| | Psychosocial development<br>University: 0.7 \| Polytechnic: 0.9 | |

When comparing study cycles (see Table 5), the results show that second cycles and integrated master's have more LO mentions than first study cycles. As displayed in Table 4, a similar pattern occurs for cognitive LOs (both in skill and knowledge outcomes). Nonetheless, integrated master's present a higher average in general skills as well as specific knowledge. In turn, generic knowledge is mentioned only by second study cycles. Results from non-cognitive LOs indicate a slight difference between study cycle, visible only in psychosocial development outcomes.

In sum, the results reveal that skill outcomes (both generic and specific) are well represented in all reports analysed (compared to knowledge outcomes). Knowledge outcomes are mostly defined as domain-specific (instead of being general in content). References to non-cognitive LOs are still modest and not representative. The major differences between the public and private sectors are the largest number of LO citations per report in the public sector, the standing of specific skills and specific knowledge in the private sector, and the relevance of non-cognitive outcomes in public institutions. The categories of LOs also differ between institution systems. Cognitive knowledge outcomes are typically mentioned by universities, whereas cognitive skills are reinforced by polytechnic institutions. Study

cycles are still a variable to consider. In general, as the qualification level increases, the number of references by each learning outcome category rises too.

**Table 5.** Incidences (average) by study cycles.

| Cognitive LOs<br>1st cycle: 5<br>2nd cycle: 6.9<br>Int. master: 6.7 | Skill<br>1st cycle: 4.25 \| 2nd cycle: 5.4<br>Integrated master's: 4.7 | Specific skills<br>1st cycle: 2.3<br>2nd cycle: 3.5<br>Integrated master's: 1.3 |
|---|---|---|
| | | Generic skills<br>1st cycle: 2.0 \| 2nd cycle:<br>1.9 \| Integrated master's: 3.33 |
| | Knowledge<br>1st cycle: = 0.75 \| 2nd cycle:<br>1.5 \| Integrated master's: 2 | Specific knowledge<br>1st cycle: 0.75 \| 2nd cycle:<br>1.2 \| Integrated master's: 2.0 |
| | | General knowledge<br>1st cycle: 0 \| 2nd cycle:<br>0.2 \| Integrated master's: 0 |
| Non-cognitive LOs<br>1st cycle: 1.8<br>2nd cycle: 1.9<br>Int. master: 1.7 | Behaviours and ethics<br>1st cycle: 1.0 \| 2nd cycle: 1.0 \| Integrated master: 1.0 | |
| | Psychosocial development<br>1st cycle: 0.8 \| 2nd cycle: 0.9 \| Integrated master: 0.7 | |

## 6. Discussion and Conclusions

The present research intents to analyse how LOs are defined by Portuguese higher education institutions, focusing on electrical or computer engineering curricula. The results were distinguished according to institutions sectors (private vs. public), institution systems (universities vs. polytechnics), and study cycles/degrees (bachelor's vs. master's vs. integrated master's).

Analysis brings into light some relevant features of electrical or computer engineering LOs. First, words such as "skills", "ability", and "knowledge" are frequently mentioned as components of LOs. Considering LOs as incorporating a wide range of theoretical and practical knowledge, skills, behaviours, and attitudes (Adam 2004; Scott 2011), the results suggest that the conceptual definition of a learning outcome is already embedded in the institutional practices. Secondly, although "knowledge" is a frequent word, content analysis indicates that cognitive skills are a central idea in LO definition. Portuguese academia assigns a clear privilege to cognitive LOs, ranging from domain-specific knowledge to the most general of reasoning and problem-solving skills (Shavelson and Huang 2003). Based on Bloom's taxonomy of educational objectives, cognitive LOs include academic skills such as application, synthesis, analysis, and evaluation. However, considering the basic distinction between knowledge and skills outcomes, the emphasis is made on skills and then on knowledge. Cognitive skills are based on complex processes of thinking, such as verbal and quantitative reasoning, information processing, comprehension, analytic operations, critical thinking, problem-solving, and the evaluation of new ideas. Both generic and specific skills are mentioned. Generic skills outcomes transcend disciplines and are transferable between diverse subject areas and contextual situations. "Solving (intellectual) problems" is frequently mentioned in the data analysed. Many LOs begin with verbs such as "understand", "(critically) analyse", "reflect", and "evaluate". Generic skills outcomes can be assessed using tests that are based on application rather than knowledge, such as the students' ability to solve intellectual problems. On the other hand, domain-specific skills are the thinking patterns used within a broad disciplinary domain, such as electrical and computer engineering. They involve an understanding of how, why, and when certain knowledge applies, not being entirely transferable throughout subject areas. An example of a domain-specific skill could be: "*Graduates will apply their knowledge of mathematics, science, engineering, technology, and computing to identify, analyse, and solve problems pertaining to design,*

*development, and implementation of electronic systems*". Considering that engineering involves the application of the principles of science and mathematics to solve world problems and innovate new products and processes across a wide range of industries and applications, it is not surprising that cognitive skills are, in fact, privileged over knowledge.

As Bloom (1956) advocates, knowledge acquisition involves the remembering, either by recognition or recall, of ideas, materials, or phenomena. The assessment of knowledge outcomes may focus either on general content knowledge or on more domain-specific knowledge. General content knowledge refers to the knowledge of a certain core curriculum whose content is considered "essential learning". An example could be: "*Understand the impact of engineering solutions in a global, economic, environmental, societal, and ethical context, including political, health, safety, manufacturability, and sustainability*".

Non-cognitive outcomes, mainly civic and ethical behaviours (Reason and Hemer 2012), are less considered. Non-cognitive development refers to changes in beliefs or the development of certain values and may be developed both through classroom instruction and out-of-class activities as supplementary activities, such as advising, tutoring, counselling, clubs, associativism, and others. Psychosocial development includes aspects of self-development such as identity development and self-esteem, interpersonal and intercultural skills, and autonomy and maturity. On the other hand, attitudes and values are closely interrelated, but attitudes are beliefs focused on a specific object, whereas values are generalized standards that transcend attitudes. The results point out social responsibility, motivation for learning, and understanding of diversity. An LO mentioned was: " *Graduates will exhibit a desire for life-long learning through higher education, technical training, teaching, membership in professional societies, and other developmental activities and will achieve positions of increased responsibility through these activities*". This gap between cognitive and non-cognitive outcomes could be particularly worrisome insofar as higher education plays a decisive role in the training of competent professionals and active, responsible, and critical-thinking citizens. In addition to acquiring rigorous technical–scientific knowledge specific to their degree, students are also expected to develop a range of transversal skills essential for a successful academic and professional career (Veiga 2022). In fact, when starting a career, many engineering graduates own the required technical knowledge but present serious behavioural mismatches (Direito et al. 2014). The lack of transversal skills, such as working well in teams and successful time management, can represent an important handicap in their careers, and the resulting limitations can significantly impair their capability to undertake the roles that the work market expects from them. Less emphasis is given to non-cognitive outcomes, mainly civic and ethical behaviours, emphasizing the need to invest in initiatives to promote the development of transversal skills during the students' university studies and alert to the importance of a regular and effective interaction between education systems and the work market.

Additionally, results provide new insights about learning outcome definitions according to different institutional sectors and systems as well as degrees. The results indicate that, despite being confined to 1000 characters, LOs are differently defined, emphasizing distinct cognitive and non-cognitive dimensions of learning. At first glance, the major differences between public and private sectors rely on the number of LO citations per report. While the private sector tends to highlight specific skills and specific knowledge, public institutions tend to give more importance to non-cognitive outcomes. Categories of LOs also differ between institutional systems. Cognitive knowledge outcomes are typically mentioned by universities, whereas cognitive skills are reinforced by polytechnic institutions, which is in line with what Portuguese legislation outlines. Different degrees are still a variable to consider. In general, as the qualification level increases, the number of references by each learning outcome category rises too. The results are aligned with the literature review. As pointed out, the knowledge, skills, and competences required in each domain of Dublin descriptors become more complex as the level of qualification increases (Joint Quality Initiative Informal Group 2004). Thus, first-cycle programs are expected to be less demanding

than second-cycle programs or an integrated master, defining more basic and narrower LOs.

Some key points can be derived from this research. Nusche typology seems to be an interesting framework to categorize LOs. The division between cognitive and non-cognitive as well as generic and specific skills and knowledge helps to identify and codify the main aspects of learning. However, if dimensions of learning are clearly defined, the criteria to assess them are still an open question. The main function of LOs seems to be merely informative, describing what knowledge and skills a student should achieve but rarely providing clues on how these achievements will be evaluated. Through the formulation of tools and mechanisms that can properly assess LOs, institutions may also be able to develop new teaching methodologies. For students, LOs can help to further understand not only what will be expected of them, in terms of knowledge and skills, but also how these will be evaluated. Assessments should reveal how well students learned what you want them to learn, while teaching/learning methodologies should ensure that they actually learn. For this to happen, assessments, learning objectives, and instructional strategies need to be closely aligned so that they are mutually reinforcing. Thus, defining assessment criteria and methods could be, as stressed by Wiliam (2010), one of the most demanding challenges higher education institutions must face.

In sum, the results provide a general overview and a more comprehensive analysis of electrical or computer engineering LOs in Portuguese higher education. In future studies, two main research vectors should be pursued. On the one hand, to enlarge the study scope, the analysis should be extended to other scientific areas. On the other hand, and from a job perspective (Kucel et al. 2016), the connection between LOs and employability skills should be explored, strengthening the bonds between academia and the labour market. LOs could be a useful tool for all learning processes, aligning students, teachers, the syllabus, and evaluations. If LOs can be assumed to pose serious risks to the successful achievement of the higher education mission, with negative repercussions not only on pedagogical and organizational fields but also for financial and social aspects; they also could/must be a useful tool for all learning processes, aligning students, teachers, the syllabus, and evaluations.

**Funding:** This research was funded by by national funds through FCT–Foundation for Science and Technology, I.P:, under Project 00757/2020.

**Institutional Review Board Statement:** The study was conducted in accordance with the Declaration of Helsinki, and approved by the Institutional Ethics Committee of Universidade Europeia (May 2020).

**Informed Consent Statement:** Informed consent was obtained from all subjects involved in the study.

**Data Availability Statement:** Data is unavailable due to privacy or ethical restrictions.

**Conflicts of Interest:** The authors declare no conflict of interest.

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
