# Peer review of "Engineering Learning Outcomes: The Possible Balance between the Passion and the Profession"

_socsci, doi:10.3390/socsci12010037_

Round 1

Reviewer 1 Report

The paper is too simplistic. There is no literature review from previous studies on LO, how they were studied and why this research is needed to advance the field. Furthermore, no theoretical framework or underpinning is provided to show the scholarship of this research. The method of analysis says that it is a qualitative study but all that is reported in the paper are numbers. The discussion is done without referring to any literature offered in the paper. The references are mostly outdated.

Author Response

First of all, I would like to thank you for reviewing the manuscript, which greatly contributed to improving its quality. So, following the opinion, a literature review of previous studies on LO was added, giving relevance to how they were studied and why this research is needed to advance the field. The description of the methodology was revised and greater emphasis was placed on reporting qualitative data. The discussion was also revised to make reference to the literature offered in the article. References have been updated.

Reviewer 2 Report

This paper uses Nusche's framework in analyzing computing-related higher education programs in Portugal. The paper can be valuable, but can definitely benefit more from providing background information and a robust presentation of results.

Section 3 started with the notion that there are multiple definitions of learning outcomes, but there was no prior discussion of this multiplicity. Maybe it would be nice to loop this back to section 1 (e.g., show how learning outcomes for similar programs but different institutions might differ, and what are the implications).

The third paragraph in section 3.1 is awkward having a single sentence only. I think it is OK to combine this as the topic sentence of the fourth paragraph.

Before diving into the Methodology, I suggest that you describe what a typical undergraduate and graduate degree program in the target domain would look like in Portugal. Is it four years? Do the students belong to a major from the get-go? What percentage of the curriculum is general education? Are the graduate programs research-heavy?

The second paragraph of the Methodology section will be easier to digest in table form or as a figure.

How about displaying the first paragraph of Results section as a word cloud?

Can you run a statistical significance test on Tables 2 to 4 raw values? Plots such as box plots or violin plots might also be useful.

It would be interesting to see examples of the learning outcomes identified. For example, was there ample mention of communication skills? For domain-specific items, how much do the institutions overlap (i.e., is a networking class common)? Do these common grounds match industry expectations of university graduates?

Can you make it a step further and give a viewpoint on how measurable are the learning outcomes presented?

A considerable value of educational policy-related papers is the actionable recommendation they can provide. What are the implications of your results?

Other things you might want to look into: T-shaped professionals, revised Bloom's Taxonomy

Minor items:

- Please add line numbers on manuscripts for review

- Typo in the abstract (possibly to be handled during the proofreading phase): This research intends to analyses the LO

- The word "Findings:" in the abstract is not necessary, unless there is some guidelines required by the journal

- Consider more inclusive language, such as using "they" for pronoun instead of the gendered "him"

- If not required by the journal, I think a period works better for the decimal points instead of a comma 

Author Response

First of all, I would like to thank you for your review, which helped a lot to improve the quality of the article. Thus, following your instructions, a better review of the literature was included, covering the previous discussion of this multiplicity of meanings of LOs. The third paragraph in section 3.1 has been revised.
Typical undergraduate and graduate programs in the target domain in Portugal have been described.
In the Methodology section, a table was created to facilitate the analysis of the presented data.
There was also a suggestion to present the results of the content analysis as a word cloud, which worked very well. Thanks for the idea.
The discussion of results has been extensively reviewed.
All Minor Items mentioned were revised, following the suggestion.

Round 2

Reviewer 2 Report

Thank you for your edits! I recognize the hard work you put into it. However, maybe because you were pressed for time, the writing ended up being hard to read. I recommend that this go through an initial round of proofreading/copyediting before moving further. I'm glad you enjoyed working with the word cloud, I like those myself! The current word cloud looks quite irregular though. Some words are repeated, which is not expected. If I may, I would like to recommend this tool which I had used before: https://www.wordclouds.com/

Author Response

Dear Sir,

Thank you very much for your kind new review.

As you suggest, a round of proofreading/copyediting was made and the word cloud was remaded (using the site pointed out).

Thanks
